# The association between smoking and cervical human papillomavirus infection among women from indigenous communities in western Botswana

**Billy M. Tsima**[1]*, **Keneilwe Motlhatlhedi**[1], **Kirthana Sharma**[2], **Patricia Rantshabeng**[1], **Andrew Ndlovu**[1], **Tendani Gaolathe**[1], **Lynnette T. Kyokunda**[1]

**1** Faculty of Medicine, University of Botswana, Gaborone, Botswana, **2** Rutgers Global Health Institute, Australia

* tsimab@ub.ac.bw

**Data Availability Statement:** All relevant data are within the manuscript and its Supporting Information files.

## Abstract

### Introduction

Cervical cancer, a malignancy caused by infection with oncogenic human papillomavirus, disproportionally affects women from low resource settings. Persistence of human papillomavirus infection may mediate an association between tobacco use and cervical cancer. In limited resource settings, women from indigenous communities are often marginalized and do not benefit from evidence-based interventions to prevent tobacco use or cervical cancer due to the limited reach of mainstream healthcare services to these communities. This study determined the association between smoking and high-risk human papillomavirus infection among women from indigenous communities in western Botswana.

### Methods

A cross-sectional study of women in indigenous communities was conducted between June and October 2022. Demographic, clinical and self-reported smoking data were collected. Cervical cytology and HPV DNA testing for high-risk human papillomavirus genotypes were performed. Multilevel multivariable logistic regression models were fit to evaluate the association between smoking and high-risk human papillomavirus infection while adjusting for potential confounders.

### Results

A total of 171 participants with a median (interquartile range) age of 40 (31–50) years from three settlements and two villages were recruited for the study. Of these, 17% were current smokers, 32.8% were living with HIV and high-risk human papillomavirus DNA was detected in 32.8% of the cervical specimens. Women who were current smokers, were nearly twice as likely to have cervical high-risk human papillomavirus infection compared to non-smokers (Adjusted Odds Ratio (95% CI); 1.74(1.09, 2.79)) after controlling for confounders.

**Funding:** The researchers would like to acknowledge the generous funding from the University of Botswana Office of Research and Development Round 39 call and grant number R1271.

**Competing interests:** The authors have declared that no competing interests exist.

## Conclusion

These data underscore the need for effective tobacco control to help mitigate cervical cancer risk in this setting. These findings can help inform decisions about targeted cervical cancer prevention and tobacco cessation interventions for women from indigenous communities.

## Introduction

Infection with the human papillomavirus (HPV) is associated with a wide range of cervical pathology including high-grade intraepithelial lesions (HSIL), a precursor to cervical cancer [1, 2]. However, the risk for development of HSIL and subsequent progression to cervical cancer differs by HPV genotype [3] with progression to cancer more likely to occur with high-risk HPV genotypes compared to the low-risk genotypes. Globally, cervical cancer ranks as the fourth leading cancer type among women of all ages [4]. Among women of reproductive age, cervical cancer is the leading cause of cancer in Southern Africa, a region with high HIV burden, cervical cancer and accounted for 23.4% of all cancers in the region in 2018 [4]. Notably, the sub-Saharan Africa (SSA) region has recorded the highest prevalence of oncogenic HPV infection, with prevalence as high as 48% in Guinea and 41% in Mozambique [5]. Furthermore, the HPV infection point prevalence in SSA is estimated to be twice as high as the global estimate at 22–24% versus 11–22% among women without cervical abnormalities [6, 7].

An association between HPV-related cervical diseases and tobacco use, a traditional risk factor for cancer, has been described [8]. This association is considered to be mediated by the higher rates of persistence of HPV infection in women who smoke tobacco [9]. In a large population based case-control study, investigators found that the risk for development of cervical cancer in situ among HPV-16 positive current smokers was 14 times higher than amongst HPV-16 negative nonsmokers. Conversely, the analysis showed that the risk among HPV-16 positive nonsmokers was only 6 times higher compared to HPV negative nonsmokers [10]. The genotoxic components of tobacco smoke that become activated to DNA-binding products in the cervical tissue has been hypothesized as the mechanism of increased risk of cervical cancer due to tobacco smoking [11]. Furthermore, tobacco smoking is known to weaken the immune system thus increasing the likelihood that HPV infection will persist [12].

Globally, indigenous populations experience a disproportionate morbidity and mortality burden secondary to tobacco use [13, 14]. Tobacco use has complex links with social determinants of health, adversely impacting populations suffering from health disparities such as indigenous communities [15]. With empirical evidence for a causal relationship between persistent HPV infection and cervical cancer now well established, effective primary and secondary prevention interventions such as regular screening (Papanicolaou (Pap) smears and assays that detect nucleic acids of the HPV), HPV vaccination and treatment of precancerous lesions are increasingly adopted in health systems across the world. However, evidence from high-income countries indicate that relative to non-indigenous women, indigenous women have higher cervical cancer morbidity and mortality rates, partly due to limited access to these interventions by indigenous women [16].

Given the significant public health burden of cervical cancer as well as tobacco use, it is important to determine the magnitude of the association between known risk factors for cervical cancer such as cervical HPV infection and tobacco smoking among women from indigenous communities. These women are often marginalized and often do not benefit from

mainstream healthcare services. Interestingly, studies exploring the intersection between HPV infection and smoking among indigenous populations have been conducted in the global north with limited contribution from other regions of the world [17–19]. There is therefore a need for data from other indigenous populations such as those from the African continent. We determined the association between high-risk human papillomavirus (hrHPV) infection and smoking among women from indigenous communities in western Botswana. These data can help inform decisions regarding targeted cervical cancer prevention and tobacco cessation interventions for women from indigenous communities.

## Methods

The current analysis follows a cross-sectional study investigating the prevalence of high-risk HPV infection among women from indigenous communities in western Botswana. The association between smoking and high-risk HPV infection was evaluated as a secondary aim. The sample size estimation was based on the prevalence aim (primary aim). Considering the high-risk HPV prevalence of 22% in Africa [20], and assuming; 1) a meaningful difference of 10% between our study population and the general population, 2) thus a prevalence of 32% among our study population, 3) statistical power of 80% and 4) type 1 error of 0.05, we needed to enrol 171 participants for the study.

Data collection began on the 6$^{th}$ of June and ended on the 18$^{th}$ of October 2022. Eligibility criteria included, women aged at least 21 years old, and residing in settlements and villages in the Kgalagadi area. Women who were currently undergoing a menstrual period, those who were pregnant or had recently delivered (less than 6 weeks), those experiencing vaginal discharge, and those with history of hysterectomy were excluded from the study. Participants were recruited through consultations with community leaders, who facilitated community consultations about the study. Potential participants were invited for face-to-face interviews, Pap smear screening, HIV and hrHPV testing conducted at health posts servicing the community. A standardized questionnaire was administered to collect sociodemographic data including self-reported smoking status. A smoker was defined as a person who smoked at least one cigarette in the past 30 days. Pap smears were undertaken by either a medical doctor or one of three study nurses with appropriate training. Pap smears were stained, preliminarily screened by a cytotechnologist and reported according to the Bethesda system [21]. Two anatomical pathologists independently reviewed the specimens, with discordant reports resolved through consensus.

### Ethical approval and consent to participate in the study

This study was approved by the University of Botswana Institutional Review Board (reference number UBR/RES/IRB/BIO/241) and issued a research permit by the Ministry of Health Research and Development Committee (reference number HPDME/13/8/1). Written informed consent to participate in the study was obtained from the participants prior to data collection. All procedures were explained in native Setswana and translated to local languages where needed.

**High risk HPV testing and genotyping.** Residual cell material from cervical smear collection brushes were preserved in a well labelled 15ml conical tube containing 10ml of phosphate buffered saline (PBS) for each participant. Cervical brushes were preserved at room temperature and transported to the University of Botswana Faculty of Health Sciences and Botswana Harvard Partnership research laboratories for testing using Atila AmpFire® Multiplex HPV Assay (AmpFIRE) manufactured by AtilaBioSystems (Mountain View, CA, USA). This is an *in vitro* nucleic acid isothermal amplification with real time fluorescence detection assay for

the qualitative genotyping of high-risk Human Papillomavirus (HPV) genotypes 16, 18, 31, 33, 35, 39, 45, 51, 52, 53, 56, 58, 59, 66, 68 from different samples. Each sample was centrifuged at 3500rpm for 5 minutes to dislodge the cells from the brush and 1ml of the cervical suspension was transferred to a 1.5 mL Eppendorf tube. Samples were concentrated by a centrifugation step, for 30 minutes at a maximum 3500 rpm. After centrifugation, the supernatant was removed completely and 30μL of the lysis buffer added to the tube. The pellet and lysis buffer were then mixed with a vortex thoroughly to resuspend the cell pellet. The resuspended pellet was transferred as a whole to the PCR tube. The PCR tubes were incubated at 95˚C for 10 minutes and used immediately after the lysis step for hrHPV detection. High-risk HPV detection was achieved by using the AmpFire HPV assay. This is an isothermal assay that used 4 primer mixes targeting the 15 hrHPV types; 16, 18, 31, 33, 35, 39, 45, 51, 52, 53, 56, 58, 59, 66 and 68. This assay has 4 channels (FAM/HEX/ROX/CY5) and specific gene targets which are fluorescently labeled and generate target amplification signals in real-time over 60 minutes. This assay also uses an internal control (IC) targeting the human-β-globin gene for each sample in the HEX channel. Lack of amplification curve in the HEX is regarded as invalid and the test has to be repeated. All samples had a detectable IC were deemed adequate for analysis.

### Data management and analysis

Data analysis was carried out using STATA version 18 (Statacorp, College Station, TX, USA). Participants' characteristics were summarized as median (interquartile range) for continuous non-normally distributed variables, while categorical variables are presented as frequencies. Smoking status was categorized as a binary variable (current smoker vs non-smoker/ex-smoker) for the main analysis. The Chi-squared test (or Fisher exact test where appropriate) as well as the Wilcoxon rank-sum test were used to compare the participants' characteristics stratified by hrHPV infection status. Considering the potential clustering effect by community, wherein women from one indigenous community are more likely to be similar to each other than they are to women in different communities, we fit a multilevel multivariable logistic regression model [22, 23] to evaluate the association between smoking status and hrHPV infection detection, adjusting for potential confounders. The binary outcome was the detection of cervical hrHPV DNA for three successive multilevel models fitted to the data. These included a null model with a random intercept at patient and community level without independent variables at any level (null model). Patient sociodemographic and clinical characteristics were selected for inclusion in a multivariable multilevel model using a cut off p-value of <0.25 at univariate level, allowing for the probability of detecting hrHPV infection to vary across communities but assuming dependence of individual explanatory variables to be similar for each community; model 1 (random intercept model). Similarly, model 2 was fit using community level variables. The final model, fitted using both individual and community level variables, introduced random coefficients to allow explanatory variable effects to vary between the indigenous communities; model 3 (random coefficient model). Crude Odds Ratio (COR) and adjusted Odds Ratio (AOR) with 95% Confidence Intervals (CI) are reported. Statistical significance was declared at a p-value <0.05 in the adjusted analyses. The intraclass correlation coefficient (ICC) was computed to assess for clustering and the assumption of the dependence of the data [24]. The ICC, Akaike Information Criteria (AIC) and the Proportional Change in Variance (PVC) were used for model comparison [25].

## Results

A total of 171 participants with a median (interquartile range) age of 40 (31–50) years from three settlements and two villages were recruited for the study. Of these, 17% were current

**Table 1. Sociodemographic and clinical characteristics of women from indigenous communities in western Botswana participating in the study.**

| Variable | Overall | hrHPV negative | hrHPV positive | p-value |
|---|---|---|---|---|
| Community; n (%) | | | | 0.001 |
| Bere | 31 (18.1) | 18(58.1) | 13(41.9) | |
| Dekar | 34 (19.9) | 21(61.8) | 13(38.2) | |
| Kacgae | 29 (17.0) | 12(41.4) | 17(58.6) | |
| Lekgwabe | 27 (15.8) | 22(81.5) | 5(18.5) | |
| Ncojane | 50 (29.2) | 42(84.0) | 8(16.0) | |
| Median age (IQR), in years | 40 (31–50) | 39(31,50) | 42(34,50) | 0.57 |
| Median parity (IQR) | 3 (2–5) | 3(2,4) | 392,5) | 0.59 |
| Smoking status; n (%) | | | | 0.001 |
| Non-smoker | 127 (24.3) | 93(80.9) | 34(60.7) | |
| Ex-smoker | 15 (8.8) | 11(9.6) | 4(7.1) | |
| Current smoker | 29 (17.0) | 11(9.6) | 18(32.1) | |
| Alcohol use; n (%) | | | | 0.08 |
| No | 74 (43.3) | 49(42.6) | 25(44.6) | |
| Stopped | 41 (24.0) | 33(28.7) | 8(14.3) | |
| Yes | 56 (32.8) | 33(28.7) | 23(41.1) | |
| Educational status; n (%) | | | | 0.56 |
| No formal education | 20 (11.8) | 12(10.5) | 8(14.3) | |
| Primary School education, incomplete | 10 (5.9) | 6(5.3) | 4(7.1) | |
| Primary School education, completed | 47 (27.7) | 28(24.6) | 19(33.9) | |
| Junior School Certificate | 64 (37.7) | 47(41.3) | 17(30.6) | |
| High School Certificate | 14 (8.2) | 11(9.7) | 3(5.4) | |
| Tertiary education | 15 (8.8) | 10(8.8) | 5(8.9) | |
| HIV status; n (%) | | | | 0.36 |
| Negative | 115 (67.3) | 80(69.6) | 35(62.5) | |
| Positive | 56 (32.8) | 35(30.4) | 21(37.5) | |
| Prior cervical cancer screening; n (%) | | | | 0.46 |
| No | 74 (43.3) | 52(45.2) | 22(39.3) | |
| Yes | 97 (56.7) | 63(54.8) | 34(60.7) | |
| Family history of cancer; n (%) | | | | 0.34 |
| No | 120 (70.2) | 78(67.8) | 42(75) | |
| Yes | 51 (29.8) | 37(32.2) | 14(25) | |
| Median (IQR) age at first intercourse, in years | 18 (16–20) | 18(16,20) | 18(17,20) | 0.79 |
| Median (IQR) number of lifetime sexual partners | 3 (2–4) | 3(2,4) | 3(2,4) | 0.92 |

IQR; Interquartile range

smokers, 32.8% were persons living with HIV and an equal proportion had hrHPV DNA detected from their cervical specimen. The sociodemographic and clinical characteristics of the participants are presented in Table 1.

Smoking status, prior cervical screening, HIV status and age were included as individual level characteristics meeting the pre-specified cut off p-value for the adjusted model. Current smokers were two times more likely to have hrHPV DNA detected from their cervical specimen compared to non-smokers; AOR (95% CI): 2.00 (1.20, 3.35) in model 1 considering individual level characteristics including HIV status. In model 2, participants from Ncojane were statistically significantly less likely to have positive cervical specimen for hrHPV DNA relative to participants from the reference community (Lekgwabe); AOR (95% CI): 0.26 (0.09, 0.75),

while there was no statistically significant difference regarding the odds of hrHPV DNA detection among participants from other communities relative to the reference community. In model 3, adjusted for both individual and community level characteristics, the association between the presence of hrHPV DNA in cervical specimen and smoking status remained statistically significantly; AOR (95% CI); 1.74(1.09, 2.79). The results of the models, associated random effects and model fitness parameters are shown in Table 2. The null model had an ICC of 0.115, indicating that 11.5% of the variance in the detection of hrHPV DNA from the participants' cervical specimen was due to inherent cluster/community variations. The final model, had the highest PCV (74.9%) indicating that both individual and community level

**Table 2. Multivariable multilevel logistic regression analysis of the association between HPV DNA detection and individual/community level characteristics of women from indigenous communities in western Botswana.**

| Variable | Category | Null Model | Model 1: Individual Level Characteristics | | Model 2: Community Level Characteristics | | Model 3: Individual and Community Level Characteristics AOR (95% CI) |
|---|---|---|---|---|---|---|---|
| | | | Crude OR (95% CI) | AOR (95% CI) | | | |
| Smoking status | Non-smoker | | 1 | 1 | | | 1 |
| | Smoker | | 2.24 (1.04,4.85) * | 2.00 (1.20,3.35) * | | | 1.74 (1.09,2.79) * |
| Prior cervical cancer screening | No | | 1 | 1 | | | 1 |
| | Yes | | 1.61 (0.78,3.29) | 1.79 (0.81,3.95) | | | 1.54 (0.71,3.36) |
| HIV status | Negative | | 1 | 1 | | | 1 |
| | Positive | | 1.69 (0.82,3.51) | 1.22 (0.55, 2.73) | | | 1.31 (0.61,2.81) |
| Age category, in years | 20–29 | | 1 | 1 | | | 1 |
| | 30–39 | | 1.20 (0.45,3.20) | 0.92(0.31, 2.74) | | | 0.91 (0.30, 2.77) |
| | 40–49 | | 2.29 (0.86,6.10) | 2.07(0.71, 6.02) | | | 2.13 (0.73,6.28) |
| | 50 or more | | 1.73 (0.60,5.00) | 0.92(0.28, 2.97) | | | 1.08 (0.33,3.54) |
| | | | | | Crude OR (95% CI) | AOR (95% CI) | |
| Community | Lekgwabe | | | | 1 | 1 | 1 |
| | Bere | | | | 0.86 (0.32,2.32) | 0.86(0.32, 2.32) | 1.16 (0.38, 3.56) |
| | Dekar | | | | 1.96 (0.70,5.48) | 1.96 (0.70,5.48) | 2.26 (0.75, 6.81) |
| | Kacgae | | | | 0.31 (0.09,1.05) | 0.31 (0.09,1.05) | 0.46 (0.12, 1.72) |
| | Ncojane | | | | 0.26 (0.09,0.75) | 0.26(0.09, 0.75)* | 0.30 (0.09, 0.95)* |
| Random effects | Variance | 0.43 | 0.29 | | 0.16 | | 0.12 |
| | ICC (%) | 11.5 | 8.2 | | 4.5 | | 3.1 |
| | PCV (%) | Ref. | 31 | | 63.4 | | 74.9 |
| | Model fitness | | | | | | |
| | AIC | 212.24 | 211.32 | | 211.07 | | 209.8 |
| | Log-likelihood | -104.12 | -97.66 | | -98.29 | | -92.42 |

ICC; Intraclass coefficient, PCV; Proportional Change in Variance; AIC; Akaike Information Criteria, AOR; Adjusted Odds Ratio. *indicates p-value <0.05.

included, explained about 75% of the variation in the detection of hrHPV DNA in the study. Model 3 had the highest log-likelihood as well as the lowest AIC indicating that this model was the best fit for the data.

## Discussion

Our study sought to evaluate the association between tobacco use (smoking) and the presence of cervical high risk HPV infection among women from indigenous communities of western Botswana. Although previous studies have demonstrated an association between tobacco smoking and cervical HPV infection, this is the first study to evaluate this association in the context of an indigenous population in sub-Saharan Africa (SSA). Women from these communities, who were current smokers by self-reports, were nearly two times more likely than non-smokers to have cervical hrHPV infection after controlling for confounders. Other investigators have also reported an association between tobacco use and HPV infection. For instance, baseline data from the multinational PApilloma TRIal against Cancer In young Adults (PATRICIA trial) showed that compared to non-smokers, women who smoked had increased odds of having HPV infection (any type) [26]. The upper end of the dose response continuum demonstrated by these investigators, is comparable to our finding of close to doubling in the odds of detecting cervical HPV DNA among smokers relative to non-smokers. Although the PATRICIA trial included a diverse group of women and its multivariable analysis adjusted for age and region (Europe, Asia Pacific, North America, Latin America, Finland), the study did not focus on indigenous women. Therefore, it is not known whether this association was also true for indigenous women in these settings, even though it may be inferred. The finding of an increased likelihood of oncogenic HPV infection among women smokers relative to non-smokers, underscores the need for greater cervical HPV related disease surveillance in women who smoke. In low- and middle-income countries, where resources may not always be available on demand to screen all women for cervical HPV related disease, consideration of smoking status could help when rationalizing the limited resources.

Prior work has demonstrated higher rates of cervical HPV infection persistence of up to 6 months among women smokers compared with non-smokers [27]. Interestingly, HPV infection persistence is known to mediate the association between smoking and cervical cancer [9]. Smokers are at increased risk of persistent HPV infection since smoking impairs both cellular responses and humoral immune responses, resulting in reduced antibody response to clear HPV infection [12]. Therefore, our finding of an almost two-fold increase in the odds of cervical hrHPV DNA detection among smokers compared to non-smokers is concerning, as these prevalent cases may signal an elevated risk for HPV infection persistence and cervical cancer in this population. Furthermore, smoking has been shown to increase the risk of subsequent HPV infection, presumably due to demonstrable impairment of the immune system [12]. Thus, the fact that close to one fifth (up to a quarter, if ex-smokers are included in the exposed group) of participants in our study were exposed to smoking, heralds a likelihood for subsequent HPV infection among these women.

However, a study from the United Kingdom showed no evidence that smoking prolonged the duration of cervical HPV infection, although, current smoking intensity was associated with increased risk for high-grade cervical intraepithelial neoplasia in that population [28]. The authors propose the influence of smoking-induced epigenetic changes as a possible explanation for any likely association between smoking and HPV acquisition.

A recent retrospective analysis evaluating the impact of HPV persistence following conization for cervical intra-epithelial neoplasia grades 2 and 3, revealed that risk for recurrent cervical lesions was higher with longer rather than shorter HPV persistence. A 12 month HPV

persistence period was associated with a 2-fold increase in 5-year cervical lesion recurrence rate compared with a 6 month HPV persistence. Interestingly, HPV persistence beyond 12 months was not associated with an increased risk for cervical lesions recurrence [29]. The investigators were not able to evaluate the impact of smoking on HPV persistence and cervical lesion recurrence within their study population. It is therefore important that future studies exploring the relationship between HPV persistence and risk of recurrence of HPV related disease post treatment, control for smoking status to better elucidate this risk. Furthermore, beyond elucidating the risk attributable to smoking for developing cervical HPV related disease among indigenous populations, future studies should also evaluate treatment outcomes relative to this risk. Interestingly, emerging data suggests comparable 10-year treatment outcomes for minimally invasive and open radical hysterectomy in low-risk early stage cervical cancer [30].

Despite the conflicting results reported in the literature, the mediating effect of smoking, whether early in the natural history of cervical HPV related disease or much later, is noteworthy. Thus, interventions aimed at reducing the burden of cervical HPV related disease should be optimized by also targeting tobacco control.

Our study revealed a relatively high prevalence of hrHPV infection, with hrHPV DNA detected in over a third (32.8%) of participants. This finding is higher than the baseline HPV prevalence of 24.2% reported by the PATRICIA trial [26] and that from a cohort of Colombian women (13.4%) [31]. Similarly, the International Agency for Research on Cancer (IARC) HPV Prevalence Surveys reported an HPV prevalence ranging from 2.5% in Spain to 26.2% in Nigeria and a pooled estimate of 12.5%, lower than our study conducted among women from indigenous communities in Botswana [32]. This relatively higher hrHPV infection in our study population compared to others may be partially explained by the fact that up to one third of our participants were living with HIV. Studies have shown that women living with HIV are three times more likely to have new HPV infection compared to HIV uninfected women [33]. Notably, participants in our study were comparable in age (median age of 40 years) to participants in these studies reporting a lower HPV prevalence. Given that acquisition of HPV infection is thought to be more likely in younger/adolescent women due to both a biological vulnerability [34, 35] and sexual risk behavior [36], our finding depicts a relatively higher prevalence in an older (as opposed to adolescents) population and thus warrants further exploration. Conversely, a higher HPV prevalence (63%) was reported among HIV negative participants of a cross-sectional study conducted in the context of planning for future HPV vaccine impact monitoring in Botswana [37]. However, this study was limited to young women at a public university aged between 18 and 22 years, who have a higher HPV acquisition risk [35] in contrast to our study population of older women.

Our data indicate a higher (17%) smoking prevalence compared to the general Botswana population (12.9%) and even higher when comparing smoking prevalence in female participants (2.7%) from the recent WHO STEPwise Approach to Non-Communicable Disease Risk Factor Surveillance (STEPS) survey [38]. The higher prevalence of smoking in this group of indigenous women when compared to the recorded national prevalence, underscores a need to prioritize these communities for targeted smoking cessation strategies. This could be an opportunity to integrate smoking cessation into cervical cancer screening strategies and vice-versa. In contrast, landmark clinical trials such as PATRICIA [26] and the Atypical Squamous Cells of Undetermined Significance Low-grade Squamous Intraepithelial lesion Triage Study (ATLS) [39], reported relatively higher smoking prevalence of 24.5% and 33% respectively compared to our study. The difference in smoking prevalence may be a reflection of the prevailing global trends wherein smoking prevalence in Africa is relatively lower compared to other continents. However, a growing body of evidence indicates an increasing parity between

smoking prevalence in SSA and that of other regions of the world as smoking rates fall in the latter but increase in Africa [40, 41]. Additionally, SSA countries represent a major target for the tobacco industry [42] and are projected to experience an increase in smoking rates due to demographic changes, economic development and tobacco product marketing [43]. This is of concern given that tobacco use is associated with a significantly increased risk of developing squamous cell cervical carcinoma [44].

Our study has some limitations worth mentioning. Notably, our analysis is based on a modest sample from Botswana and thus limits the statistical power and generalizability of our findings to other indigenous populations. Therefore, larger studies are recommended to confirm these findings. However, a bootstrap post-hoc analysis of the data with 10,000 replications yielded results comparable to those derived from the planned analysis, thus lending support to the validity of our findings. The cross-sectional design of the study limits any conclusion on causality and therefore, longitudinal studies are recommended to explore the relationship between smoking and hrHPV infection. Smoking was assessed by self-reports which may result in a social desirability bias driven underreporting and lead to exposure misclassification. However, even though self-reports are generally subjective, smoking self-reports have been shown to be accurate and widely used in the smoking research literature [45, 46].

Despite these limitations, our multilevel analytical approach is a strength. The analysis accounted for the fact that there is cultural diversity among indigenous communities and such attributes tend to cluster at the community level. Additionally, data collection, HPV testing and genotyping followed a robust plan to enhance the validity of the outcome of interest.

## Conclusion

The strong association between smoking and hrHPV infection among women from indigenous communities highlighted by our study, underscores the need for effective tobacco control in this setting in order to help mitigate against potential harm. Furthermore, the increased likelihood for having hrHPV infection among women smokers shows that smoking status can potentially aid in risk stratification for cervical HPV disease in these settings. Importantly, efforts toward tobacco control should include partnerships with members of the indigenous communities to developed tailored solutions that account for the unique challenges within these communities.

## Supporting information

**S1 Checklist. STROBE checklist of items that should be included in reports of cross-sectional case-control observational studies with detailed referencing of requirements to the text of the paper.**
(DOC)

**S1 Table. Bootstrap post-hoc analysis.**
(DOCX)

**S1 File.**
(DOCX)

**S2 File.**
(DTA)

**S3 File.**
(XLSX)

## Acknowledgments

The researchers acknowledge Mr Kesalopa from the National Health Laboratory in Gaborone, Mr Nichodimus Cooper from The Nama Heritage in Lokgwabe, Mr Leema A. Hiri from the University of Botswana San Research Center and the various community leaders from Kacgae, Bere and Dekar settlements, and Lokgwabe and Ncojane villages for their assistance. Lastly, we would like to acknowledge the participants who made this work possible.

## Author Contributions

**Conceptualization:** Billy M. Tsima, Keneilwe Motlhatlhedi, Kirthana Sharma, Patricia Rantshabeng, Tendani Gaolathe, Lynnette T. Kyokunda.

**Data curation:** Billy M. Tsima, Keneilwe Motlhatlhedi, Patricia Rantshabeng, Andrew Ndlovu, Lynnette T. Kyokunda.

**Formal analysis:** Billy M. Tsima, Keneilwe Motlhatlhedi.

**Funding acquisition:** Billy M. Tsima, Tendani Gaolathe, Lynnette T. Kyokunda.

**Investigation:** Patricia Rantshabeng, Lynnette T. Kyokunda.

**Methodology:** Billy M. Tsima, Keneilwe Motlhatlhedi, Kirthana Sharma, Andrew Ndlovu, Tendani Gaolathe, Lynnette T. Kyokunda.

**Project administration:** Kirthana Sharma, Patricia Rantshabeng, Lynnette T. Kyokunda.

**Supervision:** Tendani Gaolathe, Lynnette T. Kyokunda.

**Validation:** Andrew Ndlovu, Lynnette T. Kyokunda.

**Writing – original draft:** Billy M. Tsima, Keneilwe Motlhatlhedi, Lynnette T. Kyokunda.

**Writing – review & editing:** Billy M. Tsima, Keneilwe Motlhatlhedi, Kirthana Sharma, Patricia Rantshabeng, Andrew Ndlovu, Tendani Gaolathe, Lynnette T. Kyokunda.

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
