## [Decision Letter · Decision Letter 0]

7 Nov 2023

PONE-D-23-31632The Association Between Smoking and Cervical Human Papillomavirus Infection Among Women From Indigenous Communities in Western Botswana.PLOS ONE

Dear Dr. Tsima,

Thank you for submitting your manuscript to PLOS ONE. After careful consideration, we feel that it has merit but does not fully meet PLOS ONE’s publication criteria as it currently stands. Therefore, we invite you to submit a revised version of the manuscript that addresses the points raised during the review process.

We look forward to receiving your revised manuscript.

Kind regards,

Violante Di Donato, Ph.D,M.D.

Academic Editor

PLOS ONE

Journal Requirements:

   "The researchers would like to acknowledge the generous funding from the University of Botswana Office of Research and Development Round 39 call and grant number R1271."

Additional Editor Comments:

Dear authors,

the topic of the present article titled “ The Association Between Smoking and Cervical Human Papillomavirus Infection Among Women From Indigenous Communities in Western Botswana." is very interesting, the paper and the aim falls within the scope of the journal but the article needs major improvements.

The introduction, material and method section and tables should be modified and improved.

The manuscript should be organized better and English should be improved.

I suggest improving the manuscript with the reviewers' comments.

Reviewers' comments:

Reviewer's Responses to Questions

**Comments to the Author**

1. Is the manuscript technically sound, and do the data support the conclusions?

Reviewer #1: Partly

Reviewer #2: Partly

Reviewer #3: Yes

Reviewer #4: Yes

2. Has the statistical analysis been performed appropriately and rigorously? 

Reviewer #1: Yes

Reviewer #2: Yes

Reviewer #3: Yes

Reviewer #4: Yes

3. Have the authors made all data underlying the findings in their manuscript fully available?

Reviewer #1: No

Reviewer #2: Yes

Reviewer #3: Yes

Reviewer #4: Yes

4. Is the manuscript presented in an intelligible fashion and written in standard English?

Reviewer #1: No

Reviewer #2: Yes

Reviewer #3: Yes

Reviewer #4: Yes

5. Review Comments to the Author

Reviewer #1: Thank you for the opportunity to revise this manuscript. I think it has a great potential to advance our knowledge about the links between HPV and undesirable outcomes. A great strength of the article refers to its unique sample. Nonetheless, I would suggest the following:

- Follow the last guidelines from the STROBE statement, attaching the file and properly updating the text.

- The introduction fails to provide more details about HPV prevalence in the region in comparison to other parts of the world.

- The mechanisms underlying the association between HPV-related cervical diseases and smoking must be explored.

- The last paragraph of the introduction could include references when citing the global north. Moreover, hypotheses should be followed by references.

- Inclusion and exclusion criteria must be explicated, as well as potential loss of participants (this is also clearly required by the STROBE statement).

- Data analysis plan should be supported by references. As the sample size was quite limited, have you considered other methods (i.e., bootstrapping)?

- What was the achieved power?

- In Table 1, why no include chi-squared comparisons or other relevant methods?

- Table 2 is quite confusing. Why haven’t you presented crude OR?

- The discussion is a bit limited. In other words, if focused mostly on comparisons with other studies. I have missed more in-depth discussion about the tentative explanations for the results found in the study, including social determinants of health, health policies in the region, access to gynecological care, and others.

- Can you upload you full questionnaire?

Reviewer #2: The authors presented a study to investigate the influence of smoking on HR-HPV infections and this correlation with the risk of cervical cancer. The title summarises the content of the work appropriately. The abstract is satisfactory and the table is clear and interesting.

Although the manuscript can be considered already of high quality, I would suggest taking into account the following minor recommendations:

- I suggest another round of language revision, in order to correct a few typos and improve readability.

- Inclusion/exclusion criteria should be better clarified by extending their description.

- I find it interesting to include a reference to the long-term outcomes of patients with cervical cancer treated with a minimally-invasive approach compared to the "classic" open approach (see PMID: 37149905).

- I think that the discussion could be studied in depth. In fact, some aspects should be highlighted. In particular, the importance of cervical cancer screening and vaccination should be improved by analyzing the obstetrical outcomes and the positive margins persistence in patients. This could emphasize the importance of screening. I suggest these articles to get deeper in the topic : PMID: 33140628.

- The authors have not adequately highlighted the strengths of their study. I suggest better specifying these points.

-What are the implications of these findings for clinical practice and/or further research?

Reviewer #3: I read with great interest the Manuscript titled “The Association Between Smoking and Cervical Human Papillomavirus Infection Among Women From Indigenous Communities in Western Botswana." which falls within the aim of the Journal.

Although the manuscript can be considered already of good quality, I would suggest following recommendations:

- I suggest round of language revision, in order to correct few typos and improve readability;

- Considering topic and results if this study, I suggest that authors to add a reference to current evidence about crucial role of HPV infection persistence, also after treatment of cervical lesions. I would be glad if the authors discuss this important point PMID: 37401466 and 37747763.

Because of these reasons, the article should be revised and completed. Considering all these points, I think it could be of interest to the readers and, in my opinion, it deserves the priority to be published after minor revisions.

Reviewer #4: Sample size is small. Association could also have been studied with Chi-Square test and significance levels, along with the Models. The study is a simple cross sectional survey. To make it suitable for publication in a good journal, either increase the sample size or add other extensive variables too.

6. PLOS authors have the option to publish the peer review history of their article (what does this mean?). If published, this will include your full peer review and any attached files.

Reviewer #1: No

Reviewer #2: No

Reviewer #3: No

Reviewer #4: **Yes: **Dr. Shipra Sonkusare

---

## [Author Response · Author response to Decision Letter 0]

15 Jan 2024

Please see the attached document "Responses to Reviewers"

---

## [Decision Letter · Decision Letter 1]

2 Feb 2024

PONE-D-23-31632R1The Association Between Smoking and Cervical Human Papillomavirus Infection Among Women From Indigenous Communities in Western Botswana.PLOS ONE

Dear Dr. Tsima,

Thank you for submitting your manuscript to PLOS ONE. After careful consideration, we feel that it has merit but does not fully meet PLOS ONE’s publication criteria as it currently stands. Therefore, we invite you to submit a revised version of the manuscript that addresses the points raised during the review process.

We look forward to receiving your revised manuscript.

Kind regards,

Violante Di Donato, Ph.D,M.D.

Academic Editor

PLOS ONE

Journal Requirements:

Additional Editor Comments:

Dear authors,

the manuscript it has now been evaluated by our experts and they have recommended that minor changes be made to the submission.

Please improving the manuscript with the reviewers' comments.

Reviewers' comments:

Reviewer's Responses to Questions

**Comments to the Author**

1. If the authors have adequately addressed your comments raised in a previous round of review and you feel that this manuscript is now acceptable for publication, you may indicate that here to bypass the “Comments to the Author” section, enter your conflict of interest statement in the “Confidential to Editor” section, and submit your "Accept" recommendation.

Reviewer #1: All comments have been addressed

Reviewer #4: All comments have been addressed

2. Is the manuscript technically sound, and do the data support the conclusions?

Reviewer #1: Yes

Reviewer #4: Partly

3. Has the statistical analysis been performed appropriately and rigorously? 

Reviewer #1: Yes

Reviewer #4: No

4. Have the authors made all data underlying the findings in their manuscript fully available?

Reviewer #1: Yes

Reviewer #4: No

5. Is the manuscript presented in an intelligible fashion and written in standard English?

Reviewer #1: Yes

Reviewer #4: Yes

6. Review Comments to the Author

Reviewer #1: I am happy with the updated manuscript. The only think that is still missing is the bootstrap method. Usually, we need 10,000 resamplings. Authors just doubled their sample size. I would suggest another look at this in face of previous studies with comparable sample size.

Reviewer #4: The Chi-Square test has still not been used. It is just a cross -sectional survey and studies a basic known association. Statistical Analysis could have been stronger, and in detail.

7. PLOS authors have the option to publish the peer review history of their article (what does this mean?). If published, this will include your full peer review and any attached files.

Reviewer #1: No

Reviewer #4: **Yes: **Dr. Shipra Sonkusare

---

## [Author Response · Author response to Decision Letter 1]

21 Feb 2024

The responses to the reviewer are attached.

---

## [Decision Letter · Decision Letter 2]

28 Mar 2024

The Association Between Smoking and Cervical Human Papillomavirus Infection Among Women From Indigenous Communities in Western Botswana.

PONE-D-23-31632R2

Dear Dr. Tsima,

We’re pleased to inform you that your manuscript has been judged scientifically suitable for publication and will be formally accepted for publication once it meets all outstanding technical requirements.

Kind regards,

Violante Di Donato, Ph.D,M.D.

Academic Editor

PLOS ONE

Additional Editor Comments (optional):

The manuscript has been modified with the comments of the reviewers. It is now ready to be published.

Reviewers' comments:

Reviewer's Responses to Questions

**Comments to the Author**

1. If the authors have adequately addressed your comments raised in a previous round of review and you feel that this manuscript is now acceptable for publication, you may indicate that here to bypass the “Comments to the Author” section, enter your conflict of interest statement in the “Confidential to Editor” section, and submit your "Accept" recommendation.

Reviewer #1: All comments have been addressed

Reviewer #4: All comments have been addressed

2. Is the manuscript technically sound, and do the data support the conclusions?

Reviewer #1: Yes

Reviewer #4: Yes

3. Has the statistical analysis been performed appropriately and rigorously? 

Reviewer #1: Yes

Reviewer #4: Yes

4. Have the authors made all data underlying the findings in their manuscript fully available?

Reviewer #1: Yes

Reviewer #4: Yes

5. Is the manuscript presented in an intelligible fashion and written in standard English?

Reviewer #1: Yes

Reviewer #4: Yes

6. Review Comments to the Author

Reviewer #1: (No Response)

Reviewer #4: Its written appropriately as per the standards, though its a simple study. It would have been better to have a case control study with robust methodology.

7. PLOS authors have the option to publish the peer review history of their article (what does this mean?). If published, this will include your full peer review and any attached files.

Reviewer #1: No

Reviewer #4: **Yes: **Shipra Sonkusare

---

## [Editor Report · Acceptance letter]

15 May 2024

PONE-D-23-31632R2 

PLOS ONE

Dear Dr. Tsima, 

I'm pleased to inform you that your manuscript has been deemed suitable for publication in PLOS ONE. Congratulations! Your manuscript is now being handed over to our production team.

Kind regards, 

on behalf of

Dr. Violante Di Donato 

%CORR_ED_EDITOR_ROLE%

PLOS ONE